# Identification of Key Genes Regulating Sorghum Mesocotyl Elongation through Transcriptome Analysis

**DOI:** 10.3390/genes14061215

**Published:** 2023-06-02

**Authors:** Lan Ju, Na Lv, Feng Yin, Hao Niu, Haisheng Yan, Yubin Wang, Fangfang Fan, Xin Lv, Jianqiang Chu, Junai Ping

**Affiliations:** 1Shanxi Key Laboratory of Sorghum Genetic and Germplasm Innovation, Sorghum Research Institute, Shanxi Agricultural University, Jinzhong 030600, Chinalvxing_0_2000@126.com (X.L.);; 2College of Agriculture, Shanxi Agricultural University, Jinzhong 030600, China

**Keywords:** sorghum, mesocotyl, transcriptome analysis, key gene

## Abstract

Sorghum with longer mesocotyls is beneficialfor improving its deep tolerance, which is important for the seedling rates. Here, we perform transcriptome analysis between four different sorghum lines, with the aim of identifying the key genes regulating sorghum mesocotyl elongation. According to the mesocotyl length (ML) data, we constructed four comparison groups for the transcriptome analysis and detected 2705 common DEGs. GO and KEGG enrichment analysis showed that the most common category of DEGs were involved in cell wall, microtubule, cell cycle, phytohormone, and energy metabolism-related pathways. In the cell wall biological processes, the expression of *SbEXPA9-1*, *SbEXPA9-2*, *SbXTH25*, *SbXTH8-1*, and *SbXTH27* are increased in the sorghum lines with long ML. In the plant hormone signaling pathway, five auxin-responsive genes and eight cytokinin/zeatin/abscisic acid/salicylic acid-related genes showed a higher expression level in the long ML sorghum lines. In addition, five ERF genes showed a higher expression level in the sorghum lines with long ML, whereas two ERF genes showed a lower expression level in these lines. Furthermore, the expression levels of these genes were further analyzed using real-time PCR (RT-qPCR), which showed similar results. This work identified the candidate gene regulating ML, which may provide additional evidence to understand the regulatory molecular mechanisms of sorghum mesocotyl elongation.

## 1. Introduction

*Sorghum bicolor* (L.) Moench (sorghum) is well-known as the most drought tolerant of all crops and it is mainly cultivated in arid and semi-arid lands. In order to cope with drought stress, deep sowing is essential. However, it may lead to a low sorghum emergence rate, eventually causing a steep drop in agricultural production. Mesocotyl rapidly elongates and pushes buds out of the soil surface for seedling establishment. Additionally, it is a key factor affecting sorghum deep sowing tolerance and emergence rates in rice and maize [1,2].

The mesocotyl is a crucial organ located between the coleoptilar node and the basal part of the seminal root in sorghum seedlings [3]. The extension and division of mesocotyl cells are basic factors influencing mesocotyl elongation, and the extension of the mesocotyl cells seems more important [4]. Furthermore, selective loosening and remodeling of the cell walls is an important factor affecting mesocotyl elongation [5,6,7,8]. Expansions and xyloglucan endotransglucosylase/hydrolases (XTHs) positively regulate ML control through affecting the cell wall loosening processes [9,10].

ML is also regulated by several phytohormones [11]. Auxin promotes the elongation of mesocotyl through finely tuning the expression of polyamine oxidases (PAO), which is tightly correlated with the inhibition of cell extension in the outer tissues of maize mesocotyl [12]. Abscisic acid (ABA) promotes ML through increasing the cell division activity of meristem in rice [13]. Cytokinin (CTK) enhances ML through promoting cell division within a certain concentration, and it also antagonizes the strigolactones (SL) regulating ML [14,15]. In addition, other researchers have shown that SL, brassinosteroid (BR), ethylene (ETH), gibberellins (GAs), karrikins (KARs), and jasmonates (JAs) also regulate ML [16,17,18,19].

Up to now, only a limited number of quantitative trait loci (QTLs) or genes have been identified for ML in rice and maize. In addition, no genes were reported as regulating sorghum ML. Here, in order to illustrate the key genes affecting sorghum ML, we use transcriptome data to analyze the transcriptomic changes between two sorghum lines with long ML phenomenon and two sorghum lines with short ML phenomenon. GO and KEGG enrichment analysis showed that the most common GO and KEGG category of DEGs were involved in cell wall, microtubule, and cell cycle, phytohormone, and energy metabolism-related pathways. Then, the cell wall and plant signal transduction pathway and the ERF transcriptional factor family members were further analyzed to identify the key regulators in promoting ML. In order to verify the reliability of the transcriptome data, we further conducted qRT-PCR assays to analyze the expression level of the key genes in the sorghum lines. This work identified the candidate gene regulating ML, which may provide additional evidence to understand the regulatory molecular mechanisms of sorghum mesocotyl elongation.

## 2. Materials and Methods

### 2.1. Plant Materials and Growth Conditions

The test materials used in this study were sorghum ICSV219, TCSV361, Yidumi and Jinhui75. Yidumi and Jinhui75 are Chinese varieties. ICSV219 and TCSV361 are India varieties. The sorghum seedlings were grown in dark conditions at 28 °C.

### 2.2. Measurement of ML

To measure the ML in 4 sorghum cultivars, the sorghum seeds were sown at a depth of 2 cm in nutrient soil contained in plastic trays. The growth conditions were maintained in the dark at 28 °C. Eight days later, each sorghum seedling was carefully excavated and washed for measuring the ML with a ruler. The experiment was carried out on 3 replicates. For each replicate, at least, 6 sorghum seedlings were measured.

### 2.3. Cytological Observation of Maize Mesocotyl

Specifically, 8-day-old sorghum seedlings from the sorghum varieties were used for measuring the length of the mesocotyl cells. For each seedling, the middle part of the mesocotyl was taken for the paraffin section. Then, the paraffin sections were observed under a microscope (E100; Nikon, Tokyo, Japan) and photographed with a micro color camera (DS-U3; Nikon).

### 2.4. RNA Extraction

The total RNA in 4 different sorghum lines for the RNA-Seq and RT-qPCR assays were separately isolated from the 8-day-old sorghum mesocotyl using the TRIzol reagent (Invitrogen Life Technologies, Carlsbad, CA, USA). Additionally, the quality and integrity of the RNA were assessed using a NanoDrop spectrophotometer (Thermo Fisher Scientific, Waltham, MA, USA). Each RNA sample had 3 biological replicates for the RNA-Seq and RT-qPCR assays, respectively.

### 2.5. Library Construction, Sequencing and Sequence Analysis

The random oligonucleotides and SuperScript II were used to synthesize the first-strand cDNA. RNA sequencing was performed using the Illumina NovaSeq 6000 platform (Shanghai Personal Biotechnology Co. Ltd., Shanghai, China). The sequenced RNA data was then transformed by the software in the sequencing platform and the raw data was generated. In addition, the Cutadapt (v1.15) software was used to filter the raw data and clean, high-quality sequence data was generated for further analysis.

### 2.6. Bioinformatics Analysis

The reference genome and gene annotation files were downloaded from the genome website (https://ftp.ncbi.nlm.nih.gov/genomes/all/GCF/000/003/195/GCF_000003195.3_Sorghum_bicolor_NCBIv3, accessed on 20 December 2022 ). HISAT2 v2.0.5 was used to map the clean data to the reference genome and HTSeq (0.9.1) statistics was used to compare the read count values for each gene as the original expression of the gene. The expression was then standardized by fragments per kilobases per million reads (FPKM). An R language pheatmap (1.0.8) was carried out to perform clustering analysis of the different genes in the samples.

We mapped all the genes to the terms in the gene ontology database. Then, calculated the numbers of the differentially enriched genes for each term. In addition, we used the clusterProfiler (3.4.4) software to analyze the KEGG pathway enrichment. The prediction of the transcription factors was used to compare the plants and animals with the PlantTFDB (plant transcription factor database) to predict the transcription factor and the family information for which the transcription factor belonged.

### 2.7. RT-qPCR Assays

About 2 μg of total RNA was used for reverse transcribing to the cDNA using a reverse transcriptase kit. The qRT-PCR was performed with SYBR Premix Ex Taq (TaKaRa, Shiga, Japan) on a Light Cycler 96p (Kunpeng, Singapore). SbACTIN was used as an internal control. The experiments were performed according to 3 biological replications with similar results. All the primers used for the qRT-PCR are listed in Appendix A.

## 3. Results

### 3.1. ML Varies in Different Sorghum Lines

In this study, we selected four different sorghum lines (ICSV219, TCSV361, YIDUMI, JINHUI75) to investigate the ML. After having been grown in darkness for 8 days, the ML for these four sorghum varieties were measured. As shown in Figure 1, the ML for ICSV219 and TCSV361 were about 2–3 cm, while the ML for Yidumi and Jinhui75 were about 8–10 cm. The ML for ICSV219 and TCSV361 were extremely shorter than the ML for Yidumi and Jinhui75 (Figure 1A,B). In addition, the cell length in the middle part of the mesocotyl in the four sorghum lines showed similar results. The cell length of the mesocotyl in Yidumi and Jinhui75 were much longer than those in ICSV219 and TCSV361 (Figure 1C,D). Overall, the ML and mesocotyl cell length varied in the different sorghum lines.

### 3.2. Transcriptome Sequencing and Read Mapping

To gain insights into the molecular mechanisms underlying the ML of sorghum, we conducted transcriptome sequencing using the two sorghum lines with extremely long ML and the two sorghum lines with extremely short ML, and each with three biological replicates. A total of 12 cDNA libraries (ICSV219, TCSV361, Yidumi, Jinhui75) were constructed for the transcriptome sequencing. We obtained raw data ranging from 41.1–51.5 megabase pairs (Mbp) per library. After filtering the 12 samples for quality, clean data were obtained ranging from 38.9–48.8 megabase pairs (Mbp) per library. These reads were aligned to the sorghum bicolor reference genome, with an average rate of 97.34% unique alignment. In addition, the filtered RNA-seq data for all the samples had Q20 ratios above 98%, and Q30 ratios between 94.83% and 95.19%. These results indicated that the quality of the sequencing data was reliable and could be used for further analysis (Appendix A).

### 3.3. Identification of DEGs in Different Comparison Groups

We constructed four comparison groups, according to the ML phenomenon of these four sorghum lines (sorghum lines with long ML vs. sorghum lines with short ML). Then, DESeq (1.30.0) was carried out to analyze the difference in the gene expression. Based on the criteria of |log2FoldChange| > 1, (significant *p* value < 0.05), we identified 7541 DEGs between Yidumi and ICSV219, of which 2671 were upregulated and 4870 were downregulated; 6895 DEGs between Yidumi and TCSV361, of which 2447 were upregulated and 4448 were downregulated; 6042 DEGs between Jinhui75 and ICSV219, of which 2194 were upregulated and 3848 were down-regulated; and 6093 DEGs between Jinhui75 and TCSV361, of which 2251 were upregulated and 3842 were downregulated (Figure 2A, Appendix A). In addition, 2705 DEGs in common were detected in all comparison groups (Figure 2B).

Furthermore, to determine the functional roles of the DEGs, the functional gene categories (GO terms) were identified using gene set enrichment analysis to detect the significant biological processes. As shown in Figure 3, the most common GO category of DEGs in all the comparison groups were mainly involved in DNA replication, the cell wall, the microtubule, and the cell cycle (Figure 3, Appendix A). In addition, the Kyoto Encyclopedia for Genes and Genomes (KEGG) were identified using gene set enrichment analysis to identify the main metabolic pathways. The KEGG were classified as genetic information processing, metabolism, and environmental information processing. In the genetic information processing category, homologous recombination, mismatch repair, DNA replication, and base excision repair were all enriched in the four comparison groups. In the metabolism category, photosynthetic antenna proteins, starch and sucrose metabolism, amino sugar and nucleotide sugar metabolism, fatty acid biosynthesis, fatty acid elongation, cutin, suberin and wax biosynthesis, biotin metabolism, and brassinosteroid biosynthesis were the most common in all the comparison groups. In the environmental information processing category, the plant hormone signal transduction was enriched in all the four comparison groups (Figure 4, Appendix A).

### 3.4. Identification of the Genes Involved in the Cell Wall

As the cell wall is a key factor affecting ML, we analyzed the expression patterns of the cell wall-related gene using transcriptome analysis. Here, 102 cell wall-related DEGs were detected between Yidumi and ICSV219, of which 17 were upregulated; 53 cell wall-related DEGs were detected between Yidumi and TCSV361, of which 36 were upregulated; 78 cell wall-related DEGs were detected between Jinhui75 and ICSV219, of which 18 were upregulated; and 44 cell wall-related DEGs were detected, of which 17 were upregulated (Appendix A). In Figure 5, the heatmap shows that five cell wall loosening processes-related genes (*SbEXPA9-1*, *SbEXPA9-2*, *SbXTH25*, *SbXTH8-1,* and *SbXTH27*) were all upregulated in the sorghum varieties with long ML phenomenon compared to the sorghum varieties with short ML phenomenon (Figure 5A, Appendix A). Then, we conducted qRT-PCR assays to verify the reliability of the transcriptome data, which showed a similar expression pattern (Figure 5B–F). These results suggested that the expansion and xyloglucan endotransglucosylase/hydrolases (XTHs) genes, namely *SbEXPA9-1*, *SbEXPA9-2*, *SbXTH25*, *SbXTH8-1*, and *SbXTH27,* are key genes in promoting ML.

### 3.5. Identification of the Genes Involved in Plant Hormone Signal Transduction

Previous studies have shown that the elongation of mesocotyl was largely regulated by the action of various endogenous plant hormones, such as IAA, CK, GA3, JA, ABA, ETH, BR, and SL [17,20]. In this study, for the plant hormone signal transduction, 78, 62, 24, and 75 DEGs were identified between Yidumi/ICSV219, Yidumi/TCSV361, Jinhui75/ICSV219, and Jinhui75/TCSV361, respectively (Appendix A). In the cytokinin (CTK) signaling pathway, one two-component response regulator ARR-B family gene (*SbORR26*) and two two-component response regulator ARR-A family genes (*SbORR4*, *SbORR9*) were all upregulated in the sorghum lines with long ML phenomenon (Yidumi and Jinhui75) compared to the sorghum lines with short ML phenomenon (TCSV361 and ICSV219) (Figure 6A, Appendix A). In the auxin signaling pathway, three auxin-responsive protein genes (*SbIAA15*, *SbIAA16*, *SbIAA24*) and two small auxin-responsive genes (*SbSAUR32*, *SbSAUR50*) were all upregulated in the sorghum lines with long ML phenomenon (Yidumi and Jinhui75) compared to the sorghum lines with short ML phenomenon (TCSV361 and ICSV219) (Figure 6B). In the abscisic acid signaling pathway, the expression level of one abscisic acid receptor PYR/PYL family gene (*SbPYL8*) and two protein phosphatase 2C genes (*SbPP2C 2* and *SbPP2C 68*) in the sorghum lines with long ML phenomenon (Yidumi and Jinhui75) were higher than those in the sorghum lines with short ML phenomenon (TCSV361 and ICSV219) (Figure 6C). In the salicylic acid signaling pathway, three transcription factor TGA (*SbTGAL4*, *SbTGAL7*, *SbYGAL8*) were upregulated in n the sorghum lines with long ML. Additionally, we conducted qRT-PCR assays to verify the reliability of the transcriptome data, which showed a similar expression pattern (Appendix A). Taken together, these results illustrated that the plant hormone signal transduction affecting the sorghum ML and related genes are essential.

### 3.6. Identification of the Genes Involved in ERF Family

ERF family proteins contain a single conserved AP2/ERF domain and play a crucial role in promoting hypocotyl in *Arabidopsis* [21,22,23]. An ethylene-responsive transcription factor (*LOC_Os09g20350*) was selected as the high-confidence candidate gene for ML in rice [24]. As a result, we may assume that ERF family proteins play a vital role in regulating ML in sorghum. In order to test this idea, we analyzed the expression pattern of ERF family members in sorghum lines with long ML (Yidumi and Jinhui75) and the sorghum lines with short ML (TCSV361 and ICSV219). As shown in Figure 7A, 17 DEGs for *ERF* were detected between Yidumi and ICSV219, of which 12 were upregulated; 10 DEGs for *ERF* were detected between Yidumi and TCSV361, of which 6 were upregulated; 14 DEGs for *ERF* were detected between Jinhui75 and ICSV219, of which 5 were upregulated; and 11 DEGs for *ERF* were detected between Jinhui75 and TCSV361, of which 5 were upregulated (Figure 7A). In total, five ERF genes (*LOC8081902*, *LOC8054804*, *LOC8066410*, *LOC8073540* and *LOC8076148*) were upregulated and two ERF genes (*LOC8060355*, *LOC8065075*) were downregulated in Yidumi and Jinhui75 compared to TCSV361 and ICSV219 (Figure 7B, Appendix A). These results suggested that these seven ERF genes may be key regulators in controlling sorghum ML.

## 4. Discussion

Sorghum is mainly cultivated in arid and semi-arid lands. In order to cope with the impact of drought stress, deep sowing is an effective way to guarantee seedling rates. Previous studies have shown that sorghum varieties with long mesocotyl are beneficial to improve the sowing tolerance and emergence rates. However, up to now, there was limited knowledge on the molecular mechanism of sorghum mesocotyl elongation and no key genes involved in this process had been reported for sorghum.

In this study, we analyzed the ML in four sorghum lines and found that they showed varied phenomenon, two sorghum varieties (ICSV219 and TCSV361) with extremely short ML phenomenon and two sorghum varieties (Yidumi and Jinhui75) with extremely long ML phenomenon (Figure 1A,B). Then, we conducted transcriptome analysis for these materials and constructed four comparison groups, according to the ML phenomenon. As a result, we detected 2705 common DEGs in all the comparison groups, suggesting that these genes may be involved in regulating ML.

Previous studies have shown that many factors, such as the cell cycle, the microtubules, the cell wall, and the phytohormone affect ML [7,20,25,26,27]. Consistent with previous conclusions, in this study, GO enrichment analysis showed that the most common GO and KEGG category for the DEGs in all the comparison groups were involved in the cell wall, the microtubule and the cell cycle, and the phytohormone (Figure 3), suggesting that these factors play key roles in regulating sorghum ML. In addition, the KEGG enrichment analysis showed that the DEGs were also associated with energy metabolism-related pathways, such as starch and sucrose metabolism, amino sugar and nucleotide sugar metabolism, fatty acid biosynthesis and fatty acid elongation, which are enriched in all the comparison groups, suggesting that energy is necessary during mesocotyl growth. Finally, in order to identify the key genes in regulating ML, several pathways enriched in the GO and KEGG pathway of the transcriptome were selected and analyzed further.

Selective loosening and remodeling of the cell walls affects elongation of the mesocotyl through regulating cell elongation [5,6,7,8]. Expansins are cell wall loosening proteins that induce wall stress relaxation and volumetric extension of plant cells [9]. In rice, the overexpression of *OsEXP4* promoted elongation of the mesocotyl and coleoptile in rice [28]. In addition, XTHs, belonging to the glycoside hydrolase family 16, also control cell wall loosening processes and promote hypocotyl cell elongation in *Arabidopsis* seedlings [29]. In our study, we found that the expression of two expansin genes (*SbEXPA9-1*, *SbEXPA9-2*) and three XTHs genes (*SbXTH25*, *SbXTH8-1*, *SbXTH27*) were increased in the sorghum lines with long ML phenomenon compared to the sorghum lines with short ML phenomenon, suggesting that these five genes may promote ML by controlling the cell wall loosening processes.

Many phytohormones regulate ML. Auxin regulates multiple plant growth and developmental processes. In *Arabidopsis*, the auxin-responsive SAUR gene, *AtSAUR24,* promotes cell expansion and hypocotyls growth [30]. In our study, we identified that five auxin-responsive genes (*SbIAA15*, *SbIAA16*, *SbIAA24*, *SbSAUR32,* and *SbSAUR50*) that showed a high expression level in the long ML sorghum lines. Moreover, ABA, and CTK were reported as enhancing ML. In our study, in cytokinin and abscisic acid signaling transduction pathways, we found that five genes (*SbORR4*, *SbORR9*, *SbPYL8*, *SbPP2C 2*, *SbPP2C 68*) showed a high expression level in the sorghum lines with long ML. However, there was no study reporting that salicylic acid affects ML. Here, we found that, in the salicylic acid signaling pathway, three transcription factor TGA (*SbTGAL4*, *SbTGAL7*, *SbYGAL8*) were upregulated in the sorghum lines with long ML. These results suggested that these phytohormone-related genes may play key roles in regulating sorghum ML.

In addition, many ERF family members play a crucial role in promoting hypocotyl in *Arabidopsis*. *CBF1*, an AP2/ERF-family transcription factor, integrates light and temperature control in hypocotyl growth by promoting PIF4 and PIF5 protein abundance in the light [21]. ERF1 promotes hypocotyl elongation in the light and inhibits it in the dark [23]. ERF72 interacts directly with ARF6 and BZR1 in regulating hypocotyl elongation in *Arabidopsis*. An ethylene-responsive transcription factor (*LOC_Os09g20350*) was selected as the high-confidence candidate gene for ML in rice [24]. In our study, we assumed that five ERF genes (*LOC8081902*, *LOC8054804*, *LOC8066410*, *LOC8073540*, *LOC8076148*) may promote ML and two ERF genes (LOC8060355, LOC8065075) may inhibit ML.

## 5. Conclusions

Taken together, our transcriptome data provided a comprehensive data set for gene expressional profiles between four sorghum varieties with extreme phenomenon in ML, and identified the candidate genes that might regulate ML. Further studies are essential to functionally verify these candidate genes and investigate their role in ML regulation. Finally, the molecular makers of the key genes may be exploited in breeding to develop sorghum cultivars with long ML in the future.

## Figures and Tables

**Figure 1 genes-14-01215-f001:**
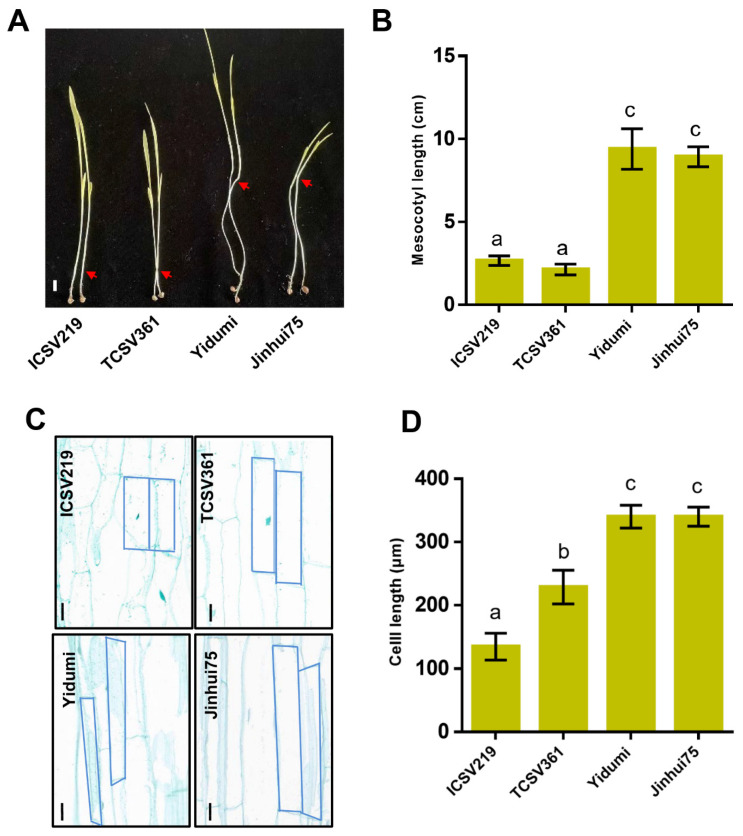
Mesocotyl length and cell length of the mesocotyl varies in the four different sorghum lines. (**A**) The mesocotyl morphology phenotype in the indicated plants. (**B**) The mesocotyl length statistics for the different plants. Different letters in the chart indicate statistically significant differences among different groups according to the one-way ANOVA analysis with Dunnett’s multiple comparison test (*p* < 0.05). (**C**) The cell length of the mesocotyl in the indicated plants. Bar = 40 um. (**D**) The cell length statistics for the mesocotyl in the different plants. Different letters in the chart indicate statistically significant differences among different groups according to the one-way ANOVA analysis with Dunnett’s multiple comparison test (*p* < 0.05).

**Figure 2 genes-14-01215-f002:**
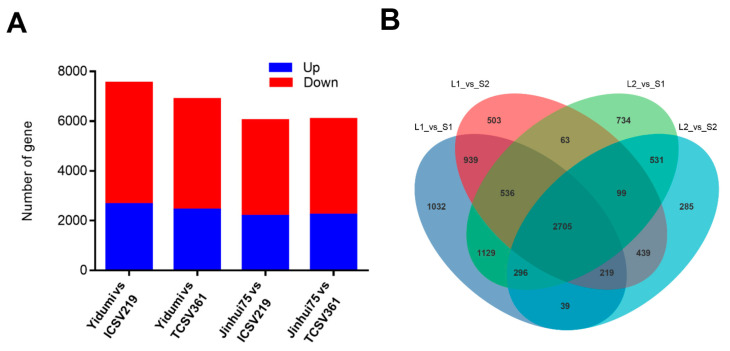
Number of DEGs in the four comparison groups. (**A**) Number of up- and downregulated DEGs among the different samples. (**B**) Venn diagrams of the DEGs in the four comparison groups. S1, S2, L1, and L2 represent ICSV219, TCSV316, Yidumi, and Jinhui75, respectively.

**Figure 3 genes-14-01215-f003:**
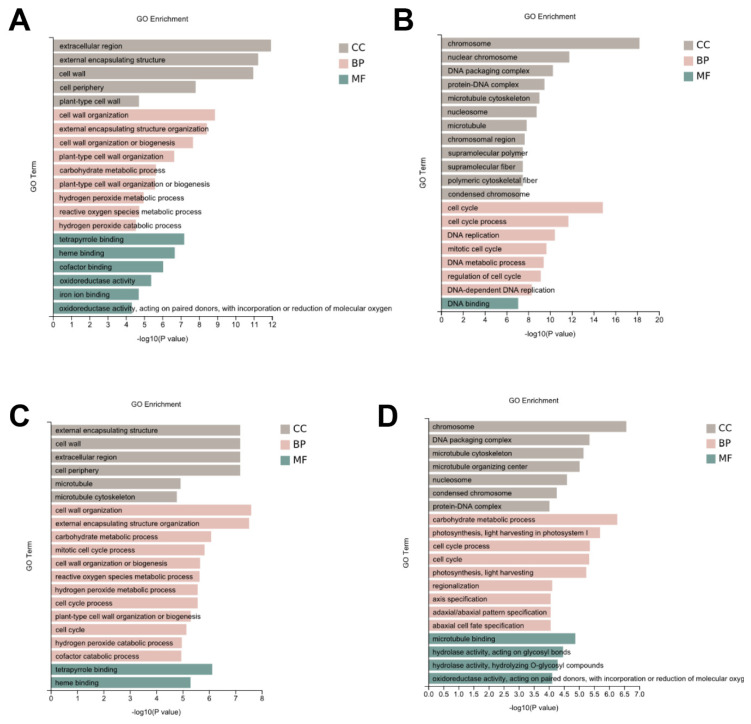
The significantly enriched GO term for the DEGs in each comparison group. The top 20 most enriched GO biological process categories among the DEGs for Yidumi vs. ICSV219 (**A**), Yidumi vs. TCSV361 (**B**), Jinhui75 vs. ICSV219 (**C**), and Jinhui75 vs. TCSV361 (**D**).

**Figure 4 genes-14-01215-f004:**
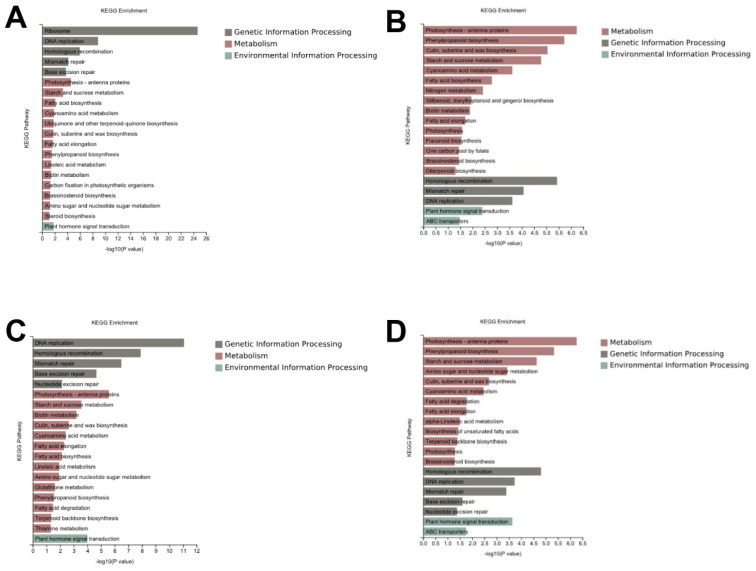
KEGG analysis of the differentially expressed genes in each comparison group. The top 20 most enriched KEGG pathway categories among the DEGs for Yidumi vs. ICSV219 (**A**), Yidumi vs. TCSV361 (**B**), Jinhui75 vs. ICSV219 (**C**), and Jinhui75 vs. TCSV361 (**D**).

**Figure 5 genes-14-01215-f005:**
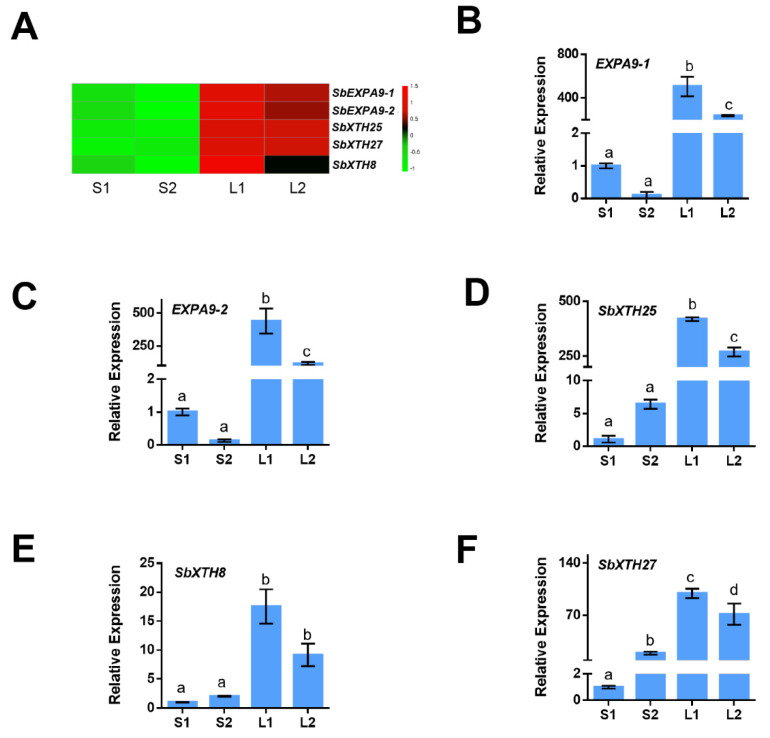
Expression pattern analysis of cell wall-related genes. (**A**) Heat map showing expression changes in key cell wall-related genes in each comparison group. (**B**–**F**) qRT-PCR analyses of cell wall-related genes. The experiments were performed three times. Error bars indicate ±SD (*n* = 3). The different lowercase letters above the bars mean significantly different at the 0.05 probability level. S1, S2, L1, and L2 represent ICSV219, TCSV361, Yidumi, and Jinhui75, respectively. Different letters in the chart indicate statistically significant differences among different groups according to the one-way ANOVA analysis with Dunnett’s multiple comparison test (*p* < 0.05).

**Figure 6 genes-14-01215-f006:**
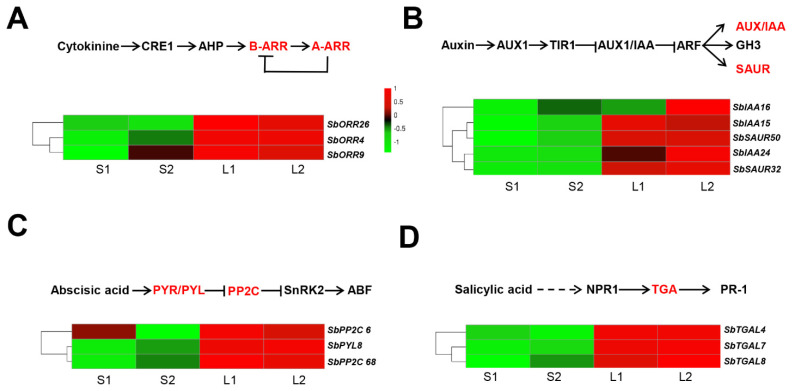
Expression pattern analysis of the hormone signaling transduction related gene. Diagram of the hormone signaling transduction pathways (upper) and heat maps (lower) of the expression of the genes related to the hormone signaling transduction of cytokinine (**A**), auxin (**B**), abscisic acid (**C**), and salicylic acid (**D**) in the four comparison groups. S1, S2, L1, and L2 represent ICSV219, TCSV361, Yidumi, and Jinhui75, respectively.

**Figure 7 genes-14-01215-f007:**
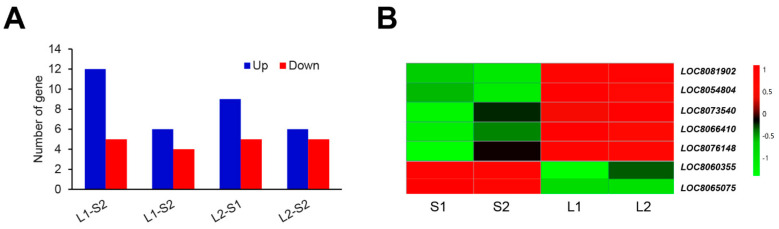
Analysis of the ERF transcription factors in each comparison group (**A**) Number of ERF DEGs in the four comparison groups. (**B**) Heat map showing the expression changes in the ERF transcription factors for each comparison group. S1, S2, L1, and L2 represent ICSV219, TCSV361, Yidumi, and Jinhui75, respectively.

## Data Availability

Not applicable.

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
