# Peer review of "Identification of Key Genes Regulating Sorghum Mesocotyl Elongation through Transcriptome Analysis"

_genes, 2023, doi:10.3390/genes14061215_

Round 1

Reviewer 1 Report

I thought it would be better to write the corrections or advice in separate paragraphs or lines, which you will find below.

45: include the complete name polyamine oxidases (PAOs)

49: this is the first time you mentioned “strigolactones (SL)”

51: karrikins(KARs)

84: 2.4 RNA Extraction

How did you check the integrity of the RNA in this experiment?

112: Which tool did you use to filter the raw reads? Include version

121: Cluster Profiler software package or library. What version?

155: It's not necessary to enter the URL again, just the version of the Sorghum genome.

161: Is it not clear how you manage to study the DEGs, what package or tool did you use?

197: In figure 5 the heatmap showed that 5 genes  controlling cell wall-loosening processes (SbEXPA9-1, SbEXPA9-2, SbXTH25, SbXTH8-1, SbXTH27) were all up-regulated in the sorghum varieties with long ML phenomenon  (Yidumi, Jinhui75) compared to the sorghum varieties with short ML phenomenon  (TCSV361, ICSV219)

210-214: Try to show the main idea without repeating too much when writing

266: As a result, we detected common 2705 DEGs in all comparison groups, suggesting that these genes may be involved in regulating ML, alongside other implicated metabolic pathways

it's fine!

Reviewer 2 Report

In the paper titled, 'Identification of key genes regulating sorghum mesocotyl elongation through transcriptome analysis', the authors describe in detail their efforts to compare differentially expressed genes between 2 sorghum lines with long mesocotyl and 2 lines with short mesocotyl. Not surprisingly, this approach identifies several candidate genes involved in cell wall regulation and hormone responses. They perform RNA-Seq as well as qPCR to validate the transcriptional responses of these genes. 

On the surface, this study is well executed and fills a knowledge gap in the literature surrounding the genes involved in sorghum mesocotyl development. However, I have 2 main concerns that prevent me from endorsing this manuscript for publication:

1. The authors overstate their findings (and/or the grammar is confusing). For example, in the abstract, the authors state, 'This work identified the candidate gene regulating ML which will help scientists breeding new varieties with longer mesocotyl'. This drastically overstates the findings as, in fact, the authors found many candidate genes that may contribute to ML length. This could be a misunderstanding; however, they write again in the Discussion line 261, '...we revealed the mechanism of ML at the transcriptional level.' This is also an overstatement of the transcriptional responses of 4 cultivars is not enough to reveal a mechanism alone. There are other instances in the discussion that border on overstatements as well. I would recommend revising the manuscript to reflect the findings more appropriately. 

2. The figure quality is very poor. The numbers inside the Venn diagram (Fig 2B) are not legible and the text in many other figures (Fig 6 & 7 for example) have small text that could easily be enlarged. 

I enjoyed reading the manuscript and look forward to your revised version. 

Many sentences were awkward to read leading to some confusion as to the authors real meaning. The grammar is almost always correct, but the word choice, verb tense, etc. do not always make sense. While I do not expect manuscripts to have perfect English, the first line of the abstract (which is difficult to understand) sets the tone for the rest of the paper. I think it is in the authors benefit for this to be revised one final time to make the writing more clear.
